# Can Liquid Biopsy Based on ctDNA/cfDNA Replace Tissue Biopsy for the Precision Treatment of EGFR-Mutated NSCLC?

**DOI:** 10.3390/jcm12041438

**Published:** 2023-02-10

**Authors:** Yi-Ze Li, Sheng-Nan Kong, Yun-Peng Liu, Yue Yang, Hong-Mei Zhang

**Affiliations:** Department of Clinical Oncology, Xijing Hospital, The Fourth Military Medical University, Xi’an 710032, China

**Keywords:** liquid biopsy, ctDNA/cfDNA, EGFR, NSCLC

## Abstract

More and more clinical trials have explored the role of liquid biopsy in the diagnosis and treatment of EGFR-mutated NSCLC. In certain circumstances, liquid biopsy has unique advantages and offers a new way to detect therapeutic targets, analyze drug resistance mechanisms in advanced patients, and monitor MRD in patients with operable NSCLC. Although its potential cannot be ignored, more evidence is needed to support the transition from the research stage to clinical application. We reviewed the latest progress in research on the efficacy and resistance mechanisms of targeted therapy for advanced NSCLC patients with plasma ctDNA EGFR mutation and the evaluation of MRD based on ctDNA detection in perioperative and follow-up monitoring.

## 1. Introduction

In recent years, the clinical application of epidermal growth factor receptor tyrosine kinase inhibitors (EGFR-TKIs) for the targeted therapy of patients with non-small-cell lung cancer (NSCLC) has been fully recognized. Treatment with EGFR-TKIs can significantly improve progression-free survival (PFS) and objective response rate (ORR) in patients with advanced EGFR-mutated NSCLC [1]. With the publication of randomized controlled clinical trials, such as ADJUVANT, ADAURA, EVIDENCE, and EVAN, more optimized targeted treatment regimens have been developed for postoperative adjuvant therapy for patients with EGFR-mutated NSCLC [2]. In addition, preoperative neoadjuvant EGFR-targeted therapy can also lead to higher disease remission rates in patients with NSCLC [3]. The best samples for EGFR testing are fresh tumor tissues and paraffin-embedded specimens, but it is difficult to obtain tumor tissues from some patients with advanced NSCLC. More and more studies have confirmed that the use of liquid biopsy to detect EGFR mutations has great potential for clinical application.

Broadly speaking, liquid biopsy is the process of testing blood or bodily secretions for cancer cells, which helps to further characterize lung cancer by identifying tumor cells or tumor DNA that are released into the blood or bodily secretions following tumor cell growth or apoptosis. As a highly sensitive technology, blood testing can not only diagnose cancer but also reveal key tumor characteristics and more comprehensive tumor genomes and dynamically monitor genetic changes during treatment. Therefore, it has great potential and broad prospects for clinical application in the field of tumor diagnosis and treatment. Currently, liquid biopsy is routinely used to detect circulating tumor cells (CTCs) [4], circulating tumor DNA (ctDNA) [5], circulating free DNA (cfDNA) [6], and exosomes [7], while research on using liquid biopsy to detect peripheral blood circulating RNAs [8], tumor-educated platelets (TEPs) [9], and circulating tumor endothelial cells (CTECs) [10] in the blood and other bodily fluids, including sputum, saliva, urine, pleural effusion (PE) [11], cerebrospinal fluid (CSF) [12], bronchoalveolar lavage fluid (BALF) [11], exhaled breath condensate (EBC) [13], etc., is gradually emerging (Figure 1). Briefly, cfDNA is a genetic material that “runs” in the blood after metabolism of almost all bodily tissues; therefore, healthy tissues, inflammatory tissues, and tumor tissues all release cfDNA into the blood. By extension, ctDNA is a fragment of cfDNA that is released into the bloodstream from primary tumors or metastatic cells. Compared to cfDNA, the amount of ctDNA in the blood is relatively low, making up only 1% of cfDNA (or even less than 0.01%) [14]. Most ctDNA fragments range from 160 to 200 bp in length and are shorter than cfDNA molecules without mutations [15,16,17]. Some studies have shown that ctDNA contains the same molecular characteristics as the tumor DNA of its origin, such as single nucleotide variation (SNV), short insertions and deletions (InDels), copy number variation (CNV) [18,19], and methylation [20,21,22]. Some studies have also found that ctDNA entering the bloodstream has a half-life of about 15 min to 2.5 h [23,24,25]. This feature ensures that ctDNA can be used as a “real-time” tumor biomarker, which can more accurately reflect tumor load than protein markers that may take weeks to present. This review introduces the progress of research on the use of ctDNA/cfDNA in the precision treatment of EGFR-mutated NSCLC, including the screening of therapeutic targets, the analysis of resistance mechanisms, and the monitoring of minimal residual disease (MRD).

## 2. Plasma ctDNA-Based EGFR Mutations Can Guide Targeted Therapy for Advanced NSCLC

Today, the treatment of advanced NSCLC has entered the era of individualized and precise targeted therapy based on molecular typing. Accurate molecular typing is the basis of this precise treatment; however, qualified tumor tissue specimens cannot be obtained from every patient, so liquid biopsy was developed. Many studies have prospectively verified the high specificity (92–100%) and positive predictive value (94.0–98.6%) of ctDNA-based EGFR mutation detection using tissues as a reference [26,27,28,29,30,31,32,33], and these two metrics have also been consistent in guiding the targeted therapy of NSCLC [30,31,34,35]. Recently, research on ctDNA-based next-generation sequencing (NGS) has demonstrated potential clinical utility [36,37,38,39]. For example, one study showed a 12.0% increase in identified actionable alterations when ctDNA-based testing was performed following tissue-based NGS testing, suggesting that ctDNA-based testing could identify additional patients with actionable genomic alterations [40]. In terms of detection timelines, Sehayek et al. compared the results and turnaround times of the molecular analysis of liquid and tissue biopsies, and found that liquid biopsies had consistent analytical results and shorter turnaround times. This means that liquid biopsy is feasible and accurate and can shorten the time required to diagnose NSCLC, especially when tumor tissues are scarce [41]. It is noteworthy that although NGS is widely touted as being more sensitive than qPCR, the available evidence suggests that all commonly used ctDNA detection techniques lack satisfactory sensitivity (e.g., qPCR, ddPCR, and NGS) [42]. A mate-analysis showed that for patients with tissue EGFR-mutated lung cancer, the therapeutic benefit of EGFR-TKIs was similar in the ctDNA+ and ctDNA− subgroups. Approximately 40% of the patients with tumor EGFR mutations that cannot be detected in ctDNA by qPCR assays at baseline could still benefit from the EGFR-TKI therapy. In such patients, repeated EGFR testing using tissue biopsies should be attempted. Of course, the impact of false negatives from the different detection methods also needs to be considered [43]. The limited yield and complex kinetics of ctDNA carry the risk of false negative results, even when using sensitive and well-validated molecular detection methods [44]. When possible, tumor tissues should be tested when ctDNA test results are negative or inconclusive. 

Table 1 presents data on the association between plasma ctDNA/cfDNA EGFR mutations and the efficacy and prognosis of targeted therapy in NSCLC patients [28,30,31,32,33,34,35,45,46,47,48,49,50,51,52,53,54,55,56,57,58]. The BENEFIT trial prospectively demonstrated the feasibility of using liquid biopsy to guide the efficacy of first-line EGFR-TKI therapy for the first time. In that study, 188 patients with EGFR mutations in their ctDNA received gefitinib treatment and showed an ORR of 72.1% with a median PFS of 9.2 months, which was basically consistent with the results of IPASS and WJTOG3405 [32]. It has further been confirmed that plasma ctDNA EGFR mutation detection is conducive to the precise targeted therapy of advanced NSCLC. Park et al. evaluated the efficacy of afatinib in treatment-naïve lung cancer patients with ctDNA-based EGFR mutations (exon 19 deletion or exon 21 point mutation). The results showed that afatinib exhibited similar ORR and PFS in lung cancer patients with EGFR mutations in their ctDNA, regardless of the outcome of the tumor EGFR mutations [35]. As a second-line treatment, osimertinib showed a favorable ORR and a median PFS for patients with plasma ctDNA T790M positivity, regardless of their tissue mutation status [45]. However, subgroup analyses of the FLAURA and AURA3 trials showed shorter median PFS for ctDNA-positive vs. ctDNA-negative patients in the osimertinib arm (FLAURA, 15.2 vs. 23.5 months; AURA3, 8.2 months vs. 10.1 months) [59,60]. This suggests that patients with tumor DNA shedding have a worse prognosis than those without tumor DNA shedding, which may, in part, be due to the lower tumor burden in ctDNA-negative patients. Park et al. also evaluated the effect of osimertinib in previously untreated metastatic NSCLC patients (*n* = 19) harboring activating EGFR mutations in their ctDNA and tumor DNA. The results showed that osimertinib had a favorable effect on such patients, with an ORR of 68% (13/19) and a median PFS of 11.1 months (95% CI = 0.0–26.7). The ORR and median PFS of ex19del (91%; 21.9 months; 95% CI = 5.5–38.3) were significantly better than those of L858R/L861Q (43%; 5.1 months; 95% CI = 2.3–7.9) [54]. Osimertinib has also demonstrated potent activity in T790M-positive NSCLC patients with CNS metastases who previously received EGFR-TKI therapy, with a median PFS of 8.4 months (95% CI = 5.8–10.9) and an ORR of 39.4%. Patients with undetectable EGFR mutations in their plasma at 6 weeks had better PFS compared to patients with detectable mutations (not reached vs. 4.5 months; 95% CI = 0.0–1.1; *p* < 0.05) [50].

Previous studies have reported that gene polymorphisms, simultaneous genomic mutations, or tumor size can affect the efficacy of EGFR-TKIs [61,62,63]. Zhou et al. first demonstrated that the abundance of EGFR mutations could predict the efficacy of EGFR-TKIs in the treatment of advanced NSCLC. Patients with low EGFR mutation abundance benefited more from EGFR-TKI treatment than did patients with wild-type EGFR, while patients with high EGFR mutation abundance benefited more from EGFR-TKI treatment than patients with low EGFR mutation abundance [64]. However, it has also been reported that the PFS for patients with low EGFR mutation abundance was similar to that for patients with wild-type EGFR. Therefore, the effect of EGFR mutation abundance on clinical outcomes in patients with advanced NSCLC, especially those with very low EGFR mutation abundance, is still unclear [65,66]. The question of whether untreated NSCLC patients with low EGFR-TKI mutation abundance detected using high-sensitivity techniques should be treated with EGFR-TKIs also remains to be determined. Wang et al. used circulating single-molecule amplification and resequencing technology (cSMART) approach to prospectively assess EGFR mutation status in plasma at the baseline and track dynamic EGFR changes during EGFR-TKI treatment. The results showed that when the cut-off EGFR mutation abundance value in plasma was 0.1%, the ORR of patients with plasma EGFR mutation abundance >0.1% was significantly higher than that of patients with plasma EGFR mutation abundance of ≤0.1% (60.0% vs. 21.4%; *p* = 0.028). The median PFS was also significantly longer in patients with a plasma EGFR mutation abundance of >0.1% compared to that of patients with a plasma EGFR mutation abundance of 0.01–0.1% (*p* = 0.0115). Additionally, one Cox multivariate analysis indicated that mutation abundance in plasma was an independent predictor of PFS (HR = 2.41, 95% CI = 1.12–5.20; *p* = 0.025) [55]. In another study, researchers used a modified semi-quantitative method (reformed-superARMS) based on ΔCt values that were generated during polymerase chain reaction (PCR) assays to investigate the impact of different baseline ctDNA EGFR mutation levels and changes in ctDNA EGFR mutations with reformed-superARMS on the outcomes and prognoses of patients receiving targeted therapy. When the ΔCt cutoff was set at 8.11 at the baseline, a significant difference in median OS was observed between the two groups (EGFR mutation ΔCt ≤ 8.11 vs. >8.11: not reached vs. 11.0 months; log rank *p* = 0.024). When the cutoff value for ΔCt was set at 4.89 after 1 month of treatment, a significant difference in median OS was again observed between the two groups (change in ΔCt > 4.89 vs. ≤4.89: not reached vs. 11.0 months; log rank *p* = 0.014) [56]. This indicates that the quantitative stratification of EGFR mutations in ctDNA can not only better select patients to be treated with EGFR-TKIs but can also help to develop better treatment strategies for patients with a low abundance of EGFR mutations.

However, the patients in the above studies had clear pathological diagnoses. The potential role of ctDNA remains unclear in patients with unknown pathological status. Deng et al. conducted a prospective study on 30 patients with suspected advanced lung cancer, in which patients with plasma EGFR-sensitizing mutations were treated with first-generation EGFR-TKIs. The results showed that EGFR-sensitizing mutations were detected in ctDNA of two-thirds of the patients. These patients received EGFR-TKI treatment, resulting in an ORR of 90% and a disease control rate (DCR) of 100%. The median PFS and OS were significantly longer than those in untreated patients (11.0 months vs. 1.0 months, *p* < 0.001) and not reached vs. 3 months (*p* < 0.001), respectively. This suggests that patients who have not been pathologically diagnosed have similar clinical outcomes and prognoses to those who have obtained histopathological diagnoses [48]. It is important to note that the amount of ctDNA entering the blood is affected by a variety of biological factors, such as tumor load and the degree of tumor vascularization, which also affect the ctDNA detection rate. For patients without an EGFR mutation in their cfDNA, test samples, test ingredients, and methods may need to be optimized. Thus, patients with poor performance (ECOG 3 to 4) and those who cannot undergo invasive procedures such as bronchoscopy or image-guided needle biopsy due to medical comorbidities or untouchable lesions could also benefit. Another similar study showed that the ORR following first-line treatment with icotinib was 52.6% (95% CI = 43.1–61.9%) for 116 patients with advanced lung cancer of unknown pathological status who had EGFR-sensitizing mutations in their ctDNA. The median PFS and OS were 10.3 months (95% CI = 8.3–12.2) and 23.2 months (95% CI = 17.7–28.0), respectively [58]. The results of these two studies suggest that ctDNA-based EGFR genotyping could help provide viable diagnoses in lung cancer patients for whom tissue biopsy is not possible. Targeted therapy for EGFR based on ctDNA assays could improve patient outcomes, but large-scale real-world exploration is still needed in the future. 

## 3. Evaluation of Resistance to EGFR-TKI Therapy Based on Plasma ctDNA Detection in NSCLC Patients

Targeted therapy has significant clinical effects on lung cancer patients with specific mutated genes, but resistance inevitably develops during treatment. Tumor heterogeneity leads to the development of multiple resistance mechanisms during targeted therapy. Therefore, the dynamic longitudinal detection of gene mutations during the treatment of NSCLC patients has become increasingly important for guiding treatment after disease progression or drug resistance has occurred. EGFR-TKI treatment resistance includes primary resistance and acquired resistance. The primary drug resistance mechanisms that have been reported so far include the presence of drug-resistant mutations in EGFR and EGFR mutations that are also combined with mutations in other related genes, such as KRAS, PIK3CA, BRAF, etc. [67,68,69]. Other non-mutated mechanisms have also been reported; for example, the insulin-like growth factor 1 receptor (IGF1R) induces primary drug resistance by interacting with EGFR [70]. The mechanisms of acquired resistance to EGFR-TKIs are mainly divided into EGFR-dependent and EGFR-independent mechanisms. Among the EGFR-dependent mechanisms, the most common cause of acquired resistance to first- and second-generation EGFR-TKIs is the EGFR T790M mutation [71,72], while the EGFR C797S mutation is considered to be the main EGFR-dependent resistance mechanism to third-generation EGFR-TKIs [73], which accounts for 10–26% of cases of resistance to second-line osimertinib treatment and approximately 7% of cases of resistance to first-line osimertinib treatment [74]. When C797S and T790M mutations are in trans, they cause resistance to third-generation EGFR TKIs but show sensitivity to a combination of first- and third-generation TKIs. On the contrary, when the mutations are in cis, they still cause resistance [75,76,77]. In addition, activation of the PI3K/Akt/mTOR pathway [73,78] and other rare mutations, such as the D761Y, T854A [79], and L747S [80] mutations, are also associated with resistance to EGFR-TKIs. EGFR-independent resistance mechanisms include alternative pathway activation and histological or phenotypic transformation. MET amplification is the most common alternative pathway activation [81,82], but other mechanisms can also lead to acquired resistance to EGFR-TKIs, including the following: HER2 amplification [83]; HER3 overexpression (HER3 activation) [84]; KRAS, BRAF, and MAPK1 mutations [85,86]; ALK fusions; GFR3-TACC3, RET-ERC1, CCDC6-RET, NTRK1-TPM3, NCOA4-RET, GOPC-ROS1, AGKBF and ESYT2-BF fusions [87,88]; the transformation of NSCLC into SCLC [82], adenocarcinoma into squamous cell carcinoma [89], and EMT [90]. 

In the process of developing drug resistance, the complexity and spatiotemporal diversity of tumor clones determine therapeutic efficacy and prognosis. The composition and competitive evolution of tumor clones have decisive impacts on therapy, which can be modulated using targeted drugs and chemotherapy. These findings could have new implications for the clinical treatment of NSCLC [91]. Liquid biopsy can identify the resistance to targeted therapy in a timely manner, showing a higher coverage of tumor heterogeneity, more complex resistance patterns, and some new resistance mutation sites, such as EGFR p.V769M and KRAS p.A11V [91]. Numerous studies have been conducted to validate the potential of non-invasive strategies for identifying the resistance mechanisms to EGFR-TKIs. Table 2 lists recent literature data on gene alterations in plasma during EGFR-TKI therapy [45,47,52,91,92,93,94,95,96,97,98,99,100,101,102,103,104,105,106,107,108,109]. More than half of patients with EGFR mutations develop EGFR T790M resistance when using first- and second-generation EGFR-TKIs [110]. The mechanistic patterns of resistance appear to differ between different first-generation TKIs (i.e., gefitinib, erlotinib, and icotinib), reflecting the heterogeneity of TKI resistance [106,111].

Papadimitrakopoulou et al. evaluated different techniques for the detection of EGFR mutations in the ctDNA of EGFR T790M-positive NSCLC patients in the AURA3 study and found that the positive percent agreement (PPA) between droplet digital polymerase chain reaction (ddPCR) and NGS was similar and that both of these methods were more sensitive than the Cobas EGFR Mutation Test v2 (Cobas plasma) [49]. However, another study showed that the abundance of T790M detected by ddPCR was significantly lower than that detected by NGS [106]. No significant differences have been found between the detection rate of EGFR T790M in tissues and that in ctDNA, and the consistency has been reported to be 50–79.4% [98,99,101,106,112]. The detection of T790M in plasma has been associated with a larger median baseline tumor size and the presence of extrathoracic diseases [49], particularly worsening bone disease [103]. Patients with EGFR ex19del have been found to be more likely to develop EGFR T790M resistance than patients with EGFR L858R (62.1% vs. 19.3%; *p* = 0.007) [91]. In particular, patients with delE746_A750 have been reported to be more likely to develop T790M mutations than patients with delS752_I759 or L858R [113]. Another study showed that female patients were more likely to develop T790M mutations than male patients [104]. Some investigators have evaluated the sensitivity of EGFR mutation screening to different cfDNA sources and their potential combinations, but unfortunately, no evidence has been reported of any improvements from the use of combinations of alternative sources, such as urine and exhaled breath condensate (EBC) [114].

Based on the results of the AURA3 trial, osimertinib was approved as the standard treatment for NSCLC patients with EGFR T790M mutations who had previously been treated with first- or second-generation EGFR-TKIs [59]. In 2016, the Cobas EGFR mutation Test v2 was approved by the US FDA for the analysis of T790M in plasma. However, for patients who have T790M detected in their plasma using this method, there has been a lack of prospective data on the clinical efficacy of osimertinib. Takahama et al. first evaluated the efficacy of osimertinib for those patients, and the results showed that the ORR of patients with palsma T790M that was detectable by the Cobas test was 55.1% (95% CI = 40.2–69.3%) and that the median PFS was 8.3 months (95% CI = 6.9–12.6 months) [52]. In fact, osimertinib can target not only the T790M mutation in tumors but also EGFR-sensitizing mutations in tumors. One study showed that both EGFR-sensitizing and T790M EGFR mutant fractions (MFs) decreased in ctDNA during osimertinib treatment, while a rebound in EGFR-sensitizing MFs was observed upon PD/treatment cessation. Significant differences were observed in ORR and PFS between the EGFR-sensitizing MF-high and EGFR-sensitizing MF-low groups at treatment cycle 4 [53]. Another study showed that patients with a low T790M relative allele frequency (RAF) in their plasma (<20%), which was calculated as the ratio of the allele frequency (AF) of T790M to the AF of sensitizing mutations, had lower ORR (0 vs. 68.8%; *p* = 0.03) and DCR (60% vs. 100%; *p* = 0.048) values when treated with osimertinib compared to patients with a high T790M RAF, suggesting that non-response to osimertinib could be due to alternative resistance mechanisms to T790M, such as MET or ERBB2 amplification and SCLC transformation. Plasma T790M RAF could also become a novel biomarker for prognostic stratification in the future [112]. Although liquid biopsy is a tool for diagnosing the presence of T790M in patients with EGFR-TKI-resistant NSCLC, tissue biopsy should be considered for patients with low T790M/activating mutation ratios in order to rule out the presence of SCLC transformation and/or other concomitant resistance mechanisms [115]. Patients whose conditions progressed in the absence of T790M mutations have been found to have worse PFS following EGFR-TKI therapy and alternative alterations, including SCLC-associated copy number changes and TP53 mutations in their plasma. These alterations are not only beneficial in tracking subsequent therapeutic responses, but they are also especially important in individuals without T790M or other known resistance mechanisms and could justify repeat-biopsies to confirm histological transformation [116]. SCLC transformation occurred in approximately 10% of patients treated with EGFR-TKIs. Such cases have been found to lose RB1 and TP53 while maintaining the original EGFR mutations, and the median OS with platinum-based chemotherapy has been reported to be 10.8 months [117]. The detection of histological transformation using ctDNA was very challenging. At present, only a few cases have been reported in which inactivated mutations in RB1 and TP53 were detected using ctDNA before the transformation of small cell lung cancer was confirmed by tissue biopsy, indicating a certain suggestive role and prompting the use of early tissue biopsy for diagnostic confirmation. In addition, genetic alterations in EGFR detected using liquid biopsy, especially EGFR amplification, have shown marked genomic instability and genome-wide hypomethylation, and these hypomethylation levels have been associated with the duration of the response to EGFR-TKI therapy [108]. 

For third-generation EGFR-TKIs (e.g., osimertinib), the most common resistance mechanism was the emergence of new C797S mutations in plasma, which have a reported detection rate of 20.0% [100]. Disease progression in osimertinib-treated patients has exhibited distinct molecular patterns, including sensitization+/T790M+/C797S+, sensitization+/T790M+/C797S−, and sensitization+/T790M−/C797S−, with median progression times of 12.27 months, 2.17 months, and 4.87 months, respectively [47]. In another study, the researchers analyzed real-world data from 56 patients with metastatic NSCLC who underwent liquid biopsies as the disease progressed. They found that T790M did not occur in patients receiving first-line osimertinib treatment, while it did occur in about 90% of patients after first- or second-generation EGFR-TKI therapy. After switching to osimertinib for second-line treatment, T790M was “lost” in 34% of these patients, and multiple EGFR and PIK3CA molecular subclones were present in their plasma ctDNA, confirming the complexity of the resistance mechanisms to osimertinib [113]. The second most common resistance mechanism was MET amplification. Patients who were resistant to third-generation EGFR-TKIs have been found to have significantly increased rates of MET amplification [109], occurring in approximately 15% of cases, compared to patients who were resistant to first- and second-generation EGFR-TKIs [118]. Moreover, the patterns of genomic alterations in patients with innate and acquired resistance to osimertinib were significantly different. Researchers have used cancer personalized profiling by deep sequencing (CAPP-seq) to analyze ctDNA and have found that PIK3CA, KRAS, or BRAF mutations and copy number gain in EGFR, ERBB2, or MET were more common in patients with innate drug resistance [119]. Some studies have reported the discovery of some rare acquired resistance mutations through ctDNA testing, such as EGFR C797G [120], TP53 R273C and KRAS G12V [106], which could impair the effectiveness of osimertinib. 

Osimertinib has potent activity against EGFR T790M-positive NSCLC with central nervous system (CNS) metastases. However, not much research has been conducted on the accuracy of the molecular analysis of CSF, especially in terms of its relationship to treatment outcomes. The results of a real-world study on NSCLC patients with CNS metastases showed that CSF was superior to plasma for detecting actionable mutations. The maximal somatic allele frequency (MSAF) was a useful bioinformatics tool for estimating the tumor fractions of cfDNA. In terms of consistency between paired CSF and plasma samples, the MSAF of CSF has been reported to be significantly higher than that of paired plasma cfDNA (*p* < 0.001). It has also been found that CSF was more able to detect changes than plasma, especially changes in copy number variations (CNV) and structural variation (SV). One study confirmed that fluid biopsies using CSF showed great potential for the identification of viable mutations and the exploration of potential resistance mechanisms in NSCLC patients with CNS metastases. However, the study did not analyze the clinical significance of genotyping based on CSF for treatment outcomes [121]. Another study also showed that plasma cfDNA did not reflect leptomeningeal metastases (LM) status very well and that CSF cfDNA was superior to plasma cfDNA for identifying mutations. The lack of effective exposure to first- and second-generation EGFR-TKIs could be one of the reasons that EGFR T790M mutation detection was less common in the CSF cfDNA of patients with meningeal metastases. However, for patients receiving the third-generation EGFR-TKI osimertinib, EGFR C797S mutations were almost evenly distributed in CSF and paired plasma [122]. Zheng et al. explored the clinical significance of paired CSF and plasma genotyping in NSCLC patients with LM for the first time, and the results showed that the identification of EGFR ex19del and T790M in CSF indicated better osimertinib efficacy (the median intracranial PFS and overall PFS were significantly longer in CSF T790M-positive patients than those in T790M-negative patients, while plasma T790M status was not associated with osimertinib efficacy). Patients who tested negative for T790M also benefited from osimertinib, but only those with EGFR ex19del or no detectable FGF3 co-mutations. Changes in the genes involved in cell cycle pathways (e.g., CDK4, CDKN2A, etc.) reduced the efficacy of osimertinib. In addition, CSF could reveal the resistance mechanisms of LM to osimertinib, such as the C797S mutation, MET dysregulation, TP53+RB1 coexistence, etc. [123].

Choi et al. evaluated whether the nanowire-based genotyping of cfDNA in CSF was useful in the treatment of LM in EGFR-mutated NSCLC. The results showed that for patients with EGFR-mutated NSCLC that progressed to LM after treatment with third-generation EGFR-TKIs, EGFR C797S was the most frequently detected mutation in CSF and its level decreased with improvements in radiation or neurological function, whereas T790M mutation levels in plasma were significantly elevated before disease progression, suggesting that nanowire technology-based CSF cfDNA genotyping could be feasible and effective for guiding LM therapy in patients with EGFR-mutated NSCLC [102].

Currently, Guardant360 CDx and FoundationOne Liquid CDx are approved by the FDA for NGS assays for gene fusion detection of DNA extracted from blood samples. However, a number of studies have shown that 5–15% of the samples tested by the DNA-NGS panel as fusion gene negative are still positive by the RNA-NGS test [124,125]. The research results of Li et al. showed that some fusion genes detected by DNA-NGS could not be used as therapeutic targets [126]. In order to solve the limitations of DNA-based fusion detection, clinical practice strategies mainly include “parallel detection” and “sequential detection” [127], but there are still many difficulties in practice. How to make more reasonable and effective use of liquid biopsy technology to explore the mechanism of EGFR-TKI resistance and guide clinical treatment still needs to be further explored.

## 4. A Marker for the New Era of NSCLC Treatment-Minimal or Molecular Residual Disease (MRD)

MRD refers to the molecular abnormalities of the origin of cancer that cannot be detected using conventional imaging (including PET/CT) or laboratory methods after treatment but can be detected by liquid biopsy and represent the persistence of lung cancer and the possibility of clinical progression. These potential sources of tumor recurrence are strongly associated with poor outcomes for patients. At present, the detection of MRD mainly relies on liquid biopsy. Using non-invasive detection methods, residual tumor lesions can provide tumor progression and specific molecular information, predict the prognosis of patients, and further guide follow-up treatment plans. In lung cancer, MRD-related research is still in the process of accumulating evidence from previous studies. The definitions of and research methods for MRD have varied greatly between different studies, and multiple ongoing trials on ctDNA-determined MRD should reveal more data regarding its value during perioperative and follow-up monitoring.

MRD is used not only to assess the risk of recurrence in patients with early-stage lung cancer but also to inform the selection of adjuvant therapy after radical local treatment (RLT) [128,129]. Since the TRACERx study in 2017 found that the dynamic detection of ctDNA in plasma could predict tumor recurrence in advance and that it had a certain correlation with the efficacy of postoperative adjuvant therapy [130], a number of subsequent studies have been carried out to explore these features further (Table 3) [131,132,133,134,135,136,137,138,139,140]. Postoperative ctDNA detection is more sensitive than imaging and can indicate disease progression or recurrence earlier. It is also helpful for auxiliary diagnosis and the formulation of follow-up treatment plans when the patient tumor burdens are low, thereby playing a key role in improving the prognoses of patients. However, the optimal monitoring time window and the exact lead time still need to be confirmed by extensive research.

Multiple studies have shown that the detection of MRD can have significant clinical relevance for guiding postoperative adjuvant therapy in NSCLC patients. Chen et al. found that ctDNA decayed rapidly after tumor resection in lung cancer patients. In general, 3 days post-surgery can be used as a benchmark for the postoperative monitoring of lung cancer. In patients who were ctDNA-positive 3 days after surgery, the recurrence-free survival (RFS) rate with adjuvant therapy has been found to be 269 days, while the RFS for those who did not receive adjuvant therapy has been found to be 111 days (*p* = 0.018). Changes in ctDNA are related to therapeutic efficacy, but further research is needed to determine whether there is a beneficial effect on OS from blood-based MRD testing [131]. Qiu et al. evaluated the predictive power of ctDNA for dynamic recurrence risk and adjuvant chemotherapy (ACT) benefit in resectable NSCLC patients. They found that ctDNA-positive patients had a significantly higher risk of recurrence compared to ctDNA-negative patients, whether in the ACT group (*p* < 0.05) or the non-ACT group (*p* < 0.05). The risk of recurrence was similar in ctDNA-negative patients, regardless of ACT use (*p* =0.46). Additionally, ctDNA-positive patients who were treated with ACT had a significantly longer RFS than ctDNA-positive patients who did not receive ACT (*p* < 0.05) [135]. A study conducted by Xia et al. also showed similar results, in that MRD-positive patients who received adjuvant therapy had improved RFS compared to those who did not receive adjuvant therapy (HR = 0.3; *p* = 0.008) [139]. Therefore, clinically high-risk NSCLC patients can be divided into two groups based on postoperative ctDNA status: ctDNA-positive patients, who were more likely to benefit from ACT therapy, and ctDNA-negative patients, who did not appear to require ACT but may benefit from slight improvements in reducing the risk of recurrence [135]. Kuang et al. found that, among patients who were ctDNA-positive after surgery, the RFS of patients who were ctDNA-positive after chemotherapy was worse than that of patients who were ctDNA-negative after chemotherapy (HR = 8.68; *p* = 0.022). For patients who were ctDNA-negative after surgery, those who were ctDNA-negative after chemotherapy had better long-term efficacy than those who were ctDNA-positive after chemotherapy (HR = 4.76; *p* = 0.047). They also found that post-chemotherapy ctDNA status was closely related to RFS and could serve as a guide for intensive postoperative treatment [133]. Another recent study investigated the association between ctDNA changes and neoadjuvant therapy (NAT) response and postoperative RFS. The results showed that ctDNA kinetics during NAT were highly consistent with pathological responses, with a sensitivity of 100%, a specificity of 83.33%, and an overall accuracy of 91.67%. Preoperatively detectable ctDNA (post-NAT) tended to be associated with worse RFS. The presence of ctDNA 3 months after surgery showed an 83% sensitivity and a 90% specificity for predicting recurrence. Molecular recurrence was better detected using ctDNA than radiography during postoperative disease surveillance, with a median time of 6.83 months. They also found that perioperative ctDNA analysis could evaluate the efficacy of NAT, reflect postoperative MRD, and predict postoperative recurrence [141].

For locally advanced or advanced lung cancer, MRD is mainly used to assess no evidence of disease (NED), i.e., complete response (CR) after treatment, oligometastatic disease (OMD) after surgery, no evidence of active disease based on existing imaging techniques, etc. [142]. CR rates in patients with advanced NSCLC after targeted therapy or immunotherapy have been reported to range from 1% to 7%. In patients with CR, monitoring for disease recurrence is a major clinical concern. The optimal duration of treatment and the availability and duration of “drug holidays” for patients receiving immune checkpoint inhibitors with durable responses remain unknown. At the same time, drug resistance is also a major challenge for patients receiving targeted therapy. In addition, approximately 30–50% of advanced lung cancers are oligometastatic at initial diagnosis [143,144,145]. Whether MRD can guide the next steps of systemic therapy when the metastatic lesions of patients with oligometastases are treated with radical therapy deserves further exploration. However, there have not been many related studies on MRD in patients with advanced NSCLC, and only a few studies have shown that the presence of ctDNA at baseline was an independent marker of poor prognosis in patients with advanced NSCLC who were receiving chemotherapy or targeted therapy. Negative ctDNA detection at the first assessment after treatment initiation has major prognostic impacts on both PFS and OS [146]. Circulating tumor DNA dynamics could predict whether patients with locally advanced NSCLC would benefit from consolidation immunotherapy [147]. For long-term responders (PFS ≥ 12 months) to PD-L1 blockade, ctDNA analysis could differentiate between those who would experience sustained benefits and those at risk of eventual progression [148]. Tang et al. performed dynamic ctDNA detection in 21 patients with oligometastatic lung cancer, 10 of whom were in the local intervention group, and the results suggested that the baseline ctDNA parameters were not associated with the PFS or OS of the patients. In the subsequent follow-up, a significant decrease in ctDNA load was observed in the local intervention group, which suggested that the local intervention could reduce the overall tumor loads of the patients. In addition, an increase in ctDNA burden was also seen in the final five patients with recurrence detected using radiography, with a mean advanced prediction time window of 6.7 months (2.9–17.9 months). Therefore, when patients with oligometastatic diseases undergo local therapy to reduce systemic tumor burden, it would be a feasible scientific hypothesis that MRD could be used to guide systemic therapy [149]. A large number of prospective studies are still needed to explore MRD-based treatment strategies for patients with locally advanced or advanced lung cancer.

Bodily fluids, such as sputum, saliva, urine, pleural effusion, and bronchoalveolar lavage fluid (BALF), could contain more tumor information than blood and reflect tumor heterogeneity in more detail. However, the emerging non-blood-derived fluid biopsy lacks consensus and clinical validation as a pretreatment and analytical method, and there are still many problems that need to be addressed. Thus, although non-blood-derived fluid biopsy could provide more opportunities to improve individualized targeted therapy and prognosis in lung cancer patients with MRD, more research is necessary to determine how best to combine information from tumor biopsy, clinical examinations, and medical imaging with genomic and MRD information from liquid biopsy [150].

## 5. Conclusions

Liquid biopsy could open up a new era for the diagnosis and treatment of lung cancer, especially when tissue biopsy cannot be performed on patients. However, more advanced molecular biological detection methods are still needed to enhance the reliability and practicality of liquid biopsy. Although liquid biopsy is not yet a substitute for tissue biopsy, with the development of high-throughput detection, early tumor screening, artificial intelligence, and other technologies, it may not merely be a dream. As a detection method for cancer, liquid biopsy would become a powerful tool that is not only consistent with pathological diagnosis but can also provide richer molecular biological information, thus offering greater potential for precise medication, dynamic detection, prognostic evaluation, and even stifling tumors in the cradle.

## Figures and Tables

**Figure 1 jcm-12-01438-f001:**
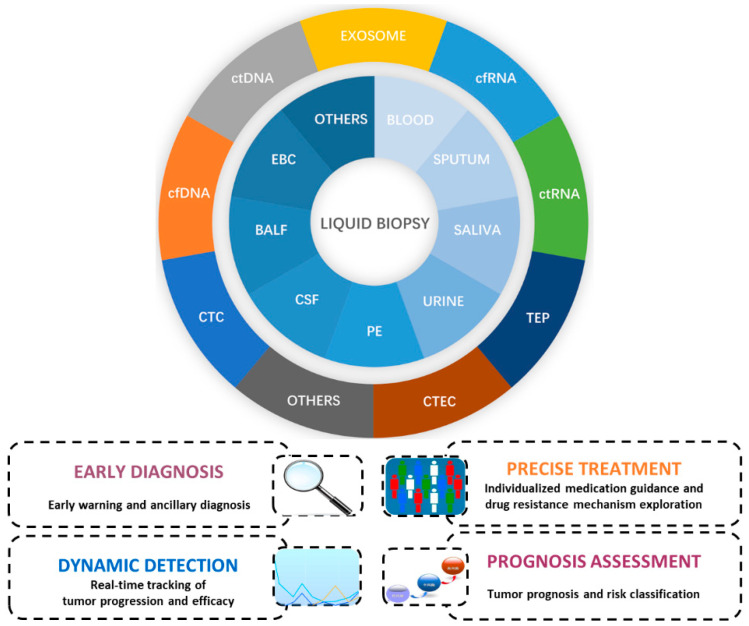
The great potential and broad prospects for the clinical application of liquid biopsy in NSCLC diagnosis and treatment. Abbreviations: PE, pleural effusion; CSF, cerebrospinal fluid; BALF, bronchoalveolar lavage fluid; EBC, exhaled breath condensate; ctDNA, circulating tumor DNA; cfDNA, circulating free DNA; CTC, circulating tumor cells; cfRNA, circulating free RNA; ctRNA, circulating tumor RNA; TEP, tumor-educated platelets; CTEC, circulating tumor endothelial cell.

**Table 1 jcm-12-01438-t001:** The correlation between plasma ctDNA/cfDNA EGFR mutations and the efficacy and prognosis of EGFR-TKI targeted therapy in patients with NSCLC.

Reference (Year)	Characteristics of Subjects	*n*	ctDNA/cfDNA Detection Method	Prognostic Relevance
[28] (2015)	➢ Stage IIIB/IV ➢ Deletion in exon 19 or L858R in exon 21 in the tumor ➢ No prior chemotherapy for metastatic diseases ➢ Treatment: erlotinib or chemotherapy	97	TaqMan assay	✓ mOS cfDNA L858R mutation vs. cfDNA exon 19 deletion: 13.7 m (95% CI = 7.1–17.7) vs. 30.0 m (95% CI = 19.3–37.7; *p* < 0.001) L858R mutation in tissues and cfDNA vs. L858R mutation in tissues but not in cfDNA: 13.7 m (95% CI = 7.1–17.7) vs. 27.7 m (95% CI = 16.1–46.2) (HR = 2.22; 95% CI = 1.09–4.52; *p* = 0.03)
[31] (2016)	➢ Group classification: Group A: positive for EGFR mutations in both tumor tissues and blood (T+/C+) (*n* = 264) Group B: positive for EGFR mutations in blood only (T−/C+) (*n* = 28) Group C: positive for EGFR mutations in tissue only (T+/C−) (*n* = 180) ➢ Treatment: EGFR-TKIs	472	ARMS, ddPCR and NGS	✓ ORR Group A: 54.6%; Group B: 46.4%; Group C: 53.9%; *p* = 0.715 ✓ mPFS Group A: 9.5 m (95% CI = 8.8–10.3); Group B: 6.5 m (95% CI = 5.1–7.9 m); Group C: 9.4 m (95% CI = 7.7–11.0 m), *p* = 0.062 ✓ mOS Group A: 25 m (95% CI = 21.7–28.4 m); Group B: 18.9 m (95% CI = 2.1–35.7 m); Group C: 29.1 m (95% CI = 24.3–34.0 m), *p* = 0.068
[45] (2018)	➢ Stage IIIB/IV ➢ Failed with prior EGFR-TKI therapy ➢ Positive for ctDNA T790M mutations ➢ Treatment: osimertinib	19	Cobas EGFR Mutaion Test v2 or PANA Mutyper	✓ ORR: 66.7% (10/15) ✓ mPFS: 8.3 m (95% CI = 7.9–8.7) ✓ mDoR: 6.8 m (95% CI = 5.3–8.3)
[46] (2018)	➢ Stage IV ➢ Activating EGFR mutations (exon 19 deletion and L858R mutation) in tumors ➢ Treatment: first-generation EGFR-TKIs	57	Qualitative (PANAMutyper) and quantitative (PANAGENE-SQI)	✓ mPFS (qualitative test) ctDNA detected vs. ctDNA undetected:11.5 m vs. 13.5 m (HR = 1.417; 95% CI = 0.80–2.52; *p* = 0.234) ✓ mPFS (quantitative test) ctDNA detected vs. ctDNA undetected: 9.8 m vs. 20.7 m (HR = 2.30; 95% CI = 1.202–4.385; *p* = 0.012)
[32] (2018)	➢ Stage IV ➢ Systemic treatment-naïve ➢ EGFR-sensitizing mutations in pre-treatment plasma ➢ Treatment: gefitinib	183	ddPCR	✓ ORR: 72.1% (132/183) ✓ mPFS: 9.5 m (95% CI = 9.07–11.04) Clear of EGFR mutations at week 8 vs. EGFR mutations persisted at week 8:11.0 m (95% CI = 9.43–12.85) vs. 2.1 m [95% CI = 1.81–3.65) (HR = 0.14; 95% CI = 0.08–0.23; *p* < 0.0001)
[47] (2019)	➢ Stage IIIB/IV ➢ EGFR mutations ➢ Failed with prior EGFR-TKI ➢ T790M mutations detected in plasma ➢ Treatment: osimertinib	22	dPCR and NGS	✓ Undetectable levels of original EGFR-sensitizing mutations after 3 months of treatment were associated with superior PFS (HR = 0.2; 95% CI = 0.05–0.7) ✓ Re-emergence of original EGFR mutations, either alone (HR = 8.8, 95% CI = 1.1–70.7) or together with T790M mutations (HR = 5.9, 95% CI = 1.2–27.9) was significantly associated with shorter PFS
[30] (2019)	➢ Stage IIIB/IV ➢ Adenocarcinoma ➢ Newly diagnosed (90.6%) or PD after EGFR-TKI therapy (3.9%) or recurrence after surgery (5.4%) ➢ Treatment: gefitinib or icotinib	71	ADx-ARMS	✓ ORR EGFR mutations in tumors (T+): 64.8% (46/71); EGFR mutations in plasma (C+): 69.0% (29/42) ✓ mPFS T+ vs. C+: 10.0 m vs. 11.0 m (*p* = 0.175) C+EGFR wild type vs. C+EGFR mutant type: 8.7 m vs. 11.0 m (*p* = 0.001)
[48] (2019)	➢ Clinically suspected advanced lung cancer ➢ Plasma EGFR sensitizing mutations ➢ Treatment: first-generation EGFR-TKIs	30	NGS or ARMS	✓ mPFS EGFR-TKIs vs. Best supportive care: 11.0 m (95% CI = 7.746–14.254) vs. 1.0 m (*p* < 0.001) ✓ mOS EGFR-TKIs vs. Best supportive care: NR vs. 3.0 m (95% CI = 2.053–3.947; *p* < 0.001)
[49] (2019)	➢ Stage III/IV ➢ EGFR T790M-positive tumors ➢ Failed with prior EGFR-TKI therapy ➢ Treatment: osimertinib vs. platinum–pemetrexed	307	Cobas EGFR Mutation Test v2	✓ mPFS T790M-negative plasma: 12.5 m with osimertinib (95% CI = 10.9-NR) ; 5.6 m with platinum–pemetrexed (95% CI = 3.2–6.7) T790M-positive plasma: 8.3 m with osimertinib (95% CI = 6.8–10.5) ; 4.2 m with platinum–pemetrexed (95% CI = 4.1–5.4)
[34] (2019)	➢ Stage IV ➢ Sensitive EGFR mutations in tumors or plasma ➢ Treatment: icotinib	66	ddPCR	✓ ORR T+: 51.51% (34/66); C+: 57.14% (24/42) ✓ DCR T+: 90.91% (60/66); C+: 92.86% (39/42)
[50] (2020)	➢ EGFR-T790M-positive tumors ➢ CNS metastases ➢ Failed with prior EGFR-TKI therapy ➢ Paired plasma and CSF samples ➢ Treatment: osimertinib	12	NGS	✓ ORR: 41.7% (5/12; 95% CI = 15.2–72.3) ✓ DCR: 83.3% (10/12; 95% CI = 51.6–97.9) ✓ mPFS: 8.3m (95% CI = 2.7-NR) Patients with undetectable plasma EGFR mutations at week 6 had better overall PFS compared to those with detectable mutations (NR vs. 4.5 m; 95% CI = 0.0–1.1; *p* < 0.05). No significant changes in PFS were observed based on the absence of detectable EGFR-sensitizing mutations in CSF at week 6 (*p* = 0.68)
[51] (2020)	➢ Metastatic EGFR-mutated lung cancer ➢ No prior treatment with EGFR-TKIs and/or VEGF inhibitors ➢ Initial detectable ctDNA ➢ Treatment: osimertinib and bevacizumab	30	ddPCR	✓ mPFS Clearance vs. no clearance (week6): 16.2 m vs. 9.8 m (*p* = 0.04) ✓ mOS Clearance vs. no clearance (week6): NR vs. 10.1 m (*p* = 0.002) ✓ ORR No association between detectable EGFR-mutated ctDNA at 6 weeks and ORR (*p* = 0.60)
[52] (2020)	➢ Advanced or recurrent NSCLC with known TKI-sensitizing EGFR mutations ➢ Failed with prior first-or second-generation EGFR-TKI therapy ➢ Positive T790M mutation in plasma ➢ Treatment: osimertinib	53	Cobas EGFR Mutation Test v2 or ddPCR	✓ ORR: 55.1% (27/49; 95% CI = 40.2–69.3%) ✓ mPFS: 8.3 m (95% CI = 6.9–12.6 m)
[53] (2020)	➢ Failed with prior first-or second-generation EGFR-TKI therapy ➢ Positive T790M mutations in plasma ➢ Treatment: osimertinib	52	Cobas EGFR Mutation Test v2, ddPCR and NGS	✓ Significant differences in ORR and PFS were observed between the sensitizing EGFR MF-high and sensitizing EGFR MF-low groups at cycle 4 (cutoff: median)
[35] (2021)	➢ Stage IIIB/IV ➢ Activating EGFR mutations in ctDNA Group A: EGFR mutation in ctDNA only (*n* = 11) Group B: EGFR mutations in ctDNA and tumor DNA (*n* = 10) ➢ Treatment: afatinib	21	PNA-based RT-PCR	✓ ORR: 74% (14/19) Group A vs. Group B: 80% (8/11) vs. 67% (6/10) (*p* = 0.35) ✓ PFS: 12.0 m Group A vs. Group B: 11.5 m vs. 12.8 m (*p* = 0.70)
[54] (2021)	➢ Stage IIIB/IV ➢ No prior exposure to EGFR-TKIs ➢ Activating EGFR mutations detected in tumor tissues or cytology specimens and ctDNA ➢ Treatment: osimertinib	19	Mutyper and Cobas EGFR Mutation Test v2	✓ ORR: 68% (13/19) ex19del vs. L858R/L861Q: 91% (10/11) vs. 43% (3/7) ✓ mPFS: 11.1 m (95%CI = 0.0–26.7) ex19del vs. L858R/L861Q: 21.9 m (95%CI = 5.5–38.3) vs. 5.1 m (95%CI = 2.3–7.9) ✓ mDoR: 17.6 m (95%CI = 3.5–31.7)
[55] (2021)	➢ Stage III/IV ➢ EGFR mutations in plasma ➢ Treatment: first generation EGFR-TKIs	54	cSMART assay	✓ ORR: 50.0% Cutoff value for plasma EGFR mutation abundance: 0.1% 60.0% for the >0.1% group vs. 21.4% for the ≤0.1% group (*p* = 0.028) ✓ mPFS >0.1% group vs. ≤0.1% group: 9.5 m vs. 5.0 m (*p* = 0.0115)
[56] (2021)	➢ Stage IIIB/IV lung adenocarcinoma ➢ Newly diagnosed or recurrence after surgery ➢ EGFR mutations in tumors ➢ R-superARMS method was used to detect the different values of EGFR mutations in plasma ➢ Treatment: first-generation EGFR-TKI monotherapy or combination therapy with antiangiogenic drugs	41	R-superARMS	✓ Baseline mOS: EGFR mutations ΔCt* ≤8.11 vs. >8.11: NR vs. 11.0 m (*p* = 0.024) ✓ 1 month after treatment mOS: mutation clearance vs. incomplete mutation clearance: NR vs. 10.4 m (*p* = 0.021) ΔCt >4.89 vs. 4.89: NR vs. 11.0 m (*p* = 0.014) mPFS: mutation clearance vs. incomplete mutation clearance: NR vs. 27.5 m (*p* = 0.088)
[57] (2022)	➢ EGFR-mutated NSCLC ➢ Treatment: first-or second-generation EGFR-TKI therapys (*n* = 14) or osimertinib (*n* = 14)	28	NGS	✓ mPFS CtDNA-clearance vs. ctDNA-non-clearance (week4): 11.4 m vs. 6.9 m (*p* = 0.091; HR = 0.42; 95% CI = 0.15–1.19) EGFR clearance vs. EGFR non-clearance (week4): 11.4 m vs. 5.67 m (*p* = 0.011; HR = 0.23; 95% CI = 0.08–0.72) Non-clearance vs. EGFR clearance only vs. total-clearance (week4): 11.4 m vs. 9.2 m vs. 5.07 m ✓ ORR Non-clearance vs. EGFR clearance only vs. total clearance: 22.2% vs. 75.0% vs. 85.7%
[58] (2022)	➢ Clinically diagnosed advanced peripheral lung cancer ➢ Systemic treatment-naive ➢ Positive pretreatment plasma EGFR-sensitizing variants ➢ Treatment: icotinib	116	SuperARMS ddPCR NGS	✓ ORR: 52.6% (95% CI, 43.1–61.9%) ✓ DCR: 84.5% (95% CI, 76.6–90.5%) ✓ mPFS: 10.3 (95% CI, 8.3–12.2) ✓ mOS: 23.2m (95% CI, 17.7–28.0)
[33] (2022)	➢ Stage IV ➢ EGFR mutations in tissues ➢ Systemic treatment-naïve ➢ Treatment: gefitinib only or gefitinib with pemetrexed and carboplatin chemotherapy	158	Cobas EGFR Mutation Test v2	✓ mPFS CtDNA negative post-treatment initiation vs. ctDNA positive: 14 m (95% CI = 12.0–17.0) vs. 8 m (95% CI = 6.0–10.0) (*p* < 0.001) ✓ mOS CtDNA-negative post-treatment initiation vs. ctDNA positive: 27 m (95% CI = 24.0–32.0 vs. 15 m (95% CI = 11.0–19.0) (*p* < 0.001)

Abbreviations: mOS, median overall survival; 95% CI, 95% confidence index; HR, hazard ratio; T+, EGFR mutations in tissues; C+, EGFR mutations in ctDNA; T−, no EGFR mutations in tissues; C−, no EGFR mutations in ctDNA; ddPCR, droplet digital PCR; NGS, next-generation sequencing; ORR, objective response rate; mPFS, median progression-free survival; DoR, duration of response; dPCR, digital PCR; ADx-ARMS, ADx-amplification refractory mutation system; NR, not reached; ctDNA, circulating tumor DNA; cfDNA, circulating free DNA; ARMS, amplification refractory mutation system; CNS, central nervous system; DCR, disease control rate; CSF, cerebrospinal fluid; VEGF, vascular endothelial growth factor. The*ΔCt value (i.e., mutant cycle threshold (Ct) value—internal control Ct value) was calculated to identify the presence of EGFR mutations, and was automatically calculated from PCR amplification fluorescence plots and the corresponding number of cycles.

**Table 2 jcm-12-01438-t002:** Literature data on the genetic alterations detected in plasma during EGFR-TKI treatment.

References (Year)	No.of Patients	Prior Treatment	Detection Method	Genetic Treatment-Resistant Alterations Detected in Plasma (%)
[106] (2021)	50	First-generation EGFR-TKIs	ddPCR and NGS	T790M: 38% (19/50)
[91] (2018)	53	First-generation EGFR-TKIs	NGS for 124-genes panel	T790M: 45.28% (24/53) EGFR point mutations: 20.75% (11/53) KRAS/NRAS point mutations: 15.09% (8/53) EGFR amplification: 7.54% (4/53) BRAF amplification: 1.8% (1/53) MET amplification: 3.7% (2/53) ERBB2 amplification: 1.8% (1/53)
[101] (2020)	37	First-generation EGFR-TKIs	ddPCRand NGS for 223-genes panel	EGFR T790M: 51.35% (19/37) TP53: 67.57% (25/37) KRAS: 8.11% (3/37) c-Met amplification: 5.41% (2/37) STK11: 5.41% (2/37) FANCA: 5.41% (2/37) ERBB2: 5.41% (2/37) PIK3CA: 2.7% (1/37) FGFR1: 2.7% (1/37) BRAF: 2.7% (1/37)
[105] (2020)	147	First-generation EGFR-TKIs	NGS for 168-genes panel	T790M: 40.13% (59/147) MET and ERBB2 amplification: 2.04% (3/147) TP53: 45.86% (61/133)
[96] (2019)	48	Icotinib	NGS for 170-genes panel	T790M: 81.2% (39/48) EGFR amplification: 72.9% (35/48) CTNNB1: 2.1% (1/48) PIK3CA: 2.1% (1/48) BRAF: 2.1% (1/48) EML4-ALK: 2.1% (1/48) SLC342-ROS1: 2.1% (1/48) Unknown mutations: 2.1% (1/48)
[45] (2018)	80	First- and second-generation EGFR-TKIs	Cobas EGFR Mutation Test v2 or PANA Mutyper	T790M: 26.3% (21/80)
[95] (2019)	66	First- and second-generation EGFR-TKIs	Cobas EGFR Mutation Test v2.	T790M: 33.3% (22/66)
[97] (2019)	50	First- and second-generation EGFR-TKIs	NGS	T790M: 71% (30/42)
[99] (2019)	118	First- and second-generation EGFR-TKIs	ARMS-PCR or super ARMS-PCR	T790M: 41.5% (49/118)
[52] (2020)	276	First- and second-generation EGFR-TKIs	Cobas EGFR Mutation Test v2 or ddPCR	T790M: 26.8% (74/276)
[103] (2020)	120	First- and second-generation EGFR-TKIs	Easy EGFR, Therascreen EGFR RGQ PCR and Cobas EGFR Mutation Test v2	T790M:25.8% (31/120)
[104] (2020)	104	First- and second-generation EGFR-TKIs	Cobas EGFR Mutation Test v2	T790M: 49% (34/104)
[94] (2018)	25	Afatinib	dPCR and NGS	T790M dPCR: 56.5% (13/23) NGS: 43.5% (10/23)
[98] (2019)	67	Afatinib	ddPCR	T790M: 73.1% (49/67)
[47] (2019)	22	First- and second-generation EGFR-TKIs	dPCR and NGS	T790M: 86% (19/22) Progression to osimertinib (16), EGFR C797S (3), A750P (1), S464L (1), amplification (1), PIK3CA E545A (3) and E545K (1)
[108] (2021)	122	First- and second-generation EGFR-TKIs	NGS for 9-genes panel	T790M: 32% (39/122) EGFR amplification: 6.6% (8/122) PIK3CA: 3.3% (4/122) MET amplification: 3.3% (4/122) HER2 amplification: 4.1% (5/122)
[102] (2020)	11	Third -generation EGFR-TKIs	Nanowire-based colorimetric cfDNA assay (EGFR mutation and MET amplification)	Plasma cfDNA profiles Drug-sensitive EGFR founder mutations: 36.3% (4/11) De novo EGFR C797S: 18.2% (2/11) MET amplification: 18.2% (2/11) EGFR T790M: 18.2% (2/11) CSF-cfDNA: Drug-sensitive EGFR founder mutations: 45.5% (5/11) De novo EGFR C797S: 36.3% (4/11) MET amplification: 18.2% (2/11) EGFR T790M: 18.2% (2/11)
[109] (2022)	49	Third-generation EGFR-TKIs	ddPCR (MET copy number gain)	MET CNG: 26.5% (13/49) MET amplification:16.3% (8/49)
[100] (2020)	26	Osimertinib	ddPCR	EGFR C797S: 20% (3/15) Loss of T790M: 33.3% (4/15)
[107] (2021)	56	Osimertinib	NGS for 39-genes panel	Second-line osimertinib (*n* = 41) EGFR C797S: 39% (16/41) Non-C797S EGFR mutations: 12% (5/41) V843I, L718Q, C724S, L792H and one patient with L718V, L718Q, L792H and G796S RB1 and TP53 inactivating mutations: 7% (3/41) EGFR amplification: 10% (4/41) MET amplification: 7% (3/41) CTNNB1 point mutations: 7% (3/41) KRAS mutations: 5% (2/41) PIK3CA activating mutations: 5% (2/41) ERBB2, PTEN, mTOR and RET mutations: 2% (1/41 each) AGK-BRAF, RET-RUFY1, TACC-FGFR3 and DLG1-BRAF fusion: 2% (1/41 each) Loss of T790M: 34% (14/41) First-line osimertinib (*n* = 7) EGFR alterations (EGFR C797S and EGFR T854A): 28.5% (2/7) MET amplification: 14.2% (1/7) EML4-ALK fusions: 14.2% (1/7) SCC/SCLC switch [RB1(R787*)]: 14.2% (1/7)
[93] (2017)	19	Osimertinib	NGS for 73-genes panel	MET amplification: 5.3% (*n* = 1) EGFR and KRAS amplification: 5.3% (*n* = 1) MEK1, KRAS or PIK3CA mutations: 5.3% (*n* = 1 each) EGFR C797S: 10.6% (*n* = 2) JAK2 mutation: 5.3% (*n* = 1) HER2 exon 20 insertion: 5.3% (*n* = 1)
[92] (2022)	-	EGFR-TKIs	NGS for 74-genes panel	First- and second-generation EGFR-TKIs (*n* = 490) EGFR T790M: 48.0% (235/490) MET amplification: 7.1% (35/490) BRAF V600E: 1.0% (5/490) KRAS mutations: 3.6% (20/490) Third-generation EGFR-TKIs (*n* = 205) MET amplification: 8.9% (16/205) EGFR C797S: 5.6% (10/205) BRAF V600E: 4.5% (8/205) KRAS mutation: 3.4% (7/205)

Abbreviations: EGFR-TKIs, epidermal growth factor receptor tyrosine kinase inhibitors; NGS, next-generation sequencing; ddPCR, droplet digital PCR; dPCR, digital PCR.

**Table 3 jcm-12-01438-t003:** The prognostic significance of circulating tumor DNA (ctDNA) detection at different time periods for resectable NSCLC.

References (Year)	Sample	Stage	Detection Methods/Study Design	Median Follow-Up Time (Month)	Detection Time	Clinical Relevance	ctDNA Positivity Precedes Radiological Recurrence by a Median Lead Time (Month)
[131] (2019)	26	I–III	NGS for a 9-gene panel/Pro	532 days for all patients and 629 days for patients who were free from progression	♦ A: immediately before surgery ♦ After tumor resection B: 5 min; C: 30 min; D: 2 h ♦ P1: 1 day post-surgery ♦ P2: 3 days post-surgery ♦ P3: 1 month post-surgery	✓ Proportion of patients who were ctDNA− positive before surgery: 17.5% (36/205) and 20.3% (36/177) ✓ Plasma ctDNA concentration showed a rapid decreasing trend after radical tumor resection. Median ctDNA half-life was 35.0 min ✓ Patients with positive MRD detection had a significantly slower ctDNA half-life than those with negative MRD detection (103.2 min vs. 29.7 min; *p* = 0.001) ✓ The RFS of patients with detectable and undetectable ctDNA concentrations at time P1 were 528 days and 543 days, respectively (*p* = 0.657) while at time P2, they were 278 days and 637 days, respectively (*p* = 0.002)	NR
[132] (2020)	20	IIA–IIIA	NGS for a 197-gene panel/Pro	12	♦ 1–2 days before surgery ♦ 3–12 days after surgery	✓ Proportion of patients who were ctDNA− positive before surgery: 40% (8/20) ✓ 8 patients (40%) were preoperatively positive for ctDNA ✓ 4 patients (20%) were preoperatively positive for ctDNA. ✓ Postoperative positivity for ctDNA also predicted shorter RFS (*p* = 0.015)	NR
[133] (2020)	38	IB–III	NGS for a 425-gene panel/Pro	15.8	♦ 1–7 days before surgery ♦ Postoperatively (within 2 weeks) ♦ After chemotherapy	✓ Proportion of patients who were ctDNA− positive before treatment: 50% (19/38) IB: 42.8%; II: 50%; III: 53.3% ✓ ctDNA was detectable in the first postoperative prechemotherapy samples of 8/35 (22.9%) patients and was associated with inferior RFS (9.6 vs. 19.6; HR = 3.69; *p* = 0.033) ✓ ctDNA was detected in the first post-chemotherapy samples of 8/36 (22.2%) patients and was also associated with inferior RFS (9.6 vs. NR; HR = 8.76; *p* < 0.001)	NR
[134] (2021)	174	I–III	ARMs for EGFR/Pro	NA	♦ 1 day before surgery	✓ Proportion of patients with ctDNA EGFR mutations before surgery: 15.5% (27/174) ✓ The overall 5-year survival rates for patients with ctDNA EGFR mutations and those without ctDNA EGFR mutations were 18.5% and 76.9%, respectively. ✓ For patients with ctDNA EGFR mutations, the median OS and DFS were 29.00 ± 2.55 and 19.00 ± 2.50 months, respectively, which were both significantly worse than those of patients without ctDNA EGFR mutations (*p* < 0.001) ✓ ctDNA EGFR mutations were an independent risk factor of OS (HR = 3.289; 95% CI = 1.816–5.956; *p* < 0.001) and DFS (HR = 4.860, 95% CI = 2.660–8.880, *p* < 0.001)	NR
[135] (2021)	116	I–IV	NGS for a 139-gene panel/Pro	NA	♦ Before surgery ♦ 1 month post-surgery ♦ Post-ACT ♦ Longitudinal detection	✓ Proportion of patients who were ctDNA− positive before surgery: 69.3% (61/88) I/II: 61.0% (25/41); III: 76.1% (35/46) Proportion of patients who were ctDNA-positive after surgery: 21.2% (18/85); after the completion of ACT: 12.5% (8/64) ✓ Both postsurgical (*p* < 0.001) and post-ACT (*p* < 0.05) ctDNA positivity were associated with worse recurrence-free survival rates ✓ In stage II-III patients, those who were ctDNA-positive after surgery benefited from ACT (*p* < 0.05)	2.93
[136] (2021)	77	I–IV	cSMART for a 127-gene panel/Pro	46	♦ 1–7 days before surgery ♦ Longitudinal detection	✓ Proportion of patients who were ctDNA-positive before surgery: 59.7% (46/77) I: 43.9% (18/41); II: 72.2% (13/18); III: 81.3% (13/16); IV:100% (2/2) ✓Proportion of patients who were ctDNA-positive post-surgery: 42.25% (30/71) I: 29.0% (11/38); II: 41.2% (7/17); III: 71.4% (10/14); IV: 100% (2/2) ✓ Patients with higher stage (III/IV) cancers and preoperative ctDNA-positive status demonstrated significant risks for recurrence and death, (2.8–3.4-fold risk and 3.8–4.0-fold risk, respectively) ✓ Preoperative ctDNA-positive patients were associated with lower RFS (HR = 3.812; *p* = 0.0005) and OS (HR = 5.004; *p* = 0.0009) ✓ Postoperative ctDNA-positive patients were also associated with lower RFS (HR = 3.076; *p*= 0.0015) and OS (HR = 3.195; *p* = 0.0053) ✓ Disease recurrence occurred in 63.3% (19/30) of postoperative ctDNA-positive patients	12.6
[137] (2021)	119	I–IIIA	NGS for a 425-gene panel/Pro	30.7	♦ 1 week before surgery ♦ 1 month after surgery ♦ Longitudinal detection	✓ Preoperative ctDNA was detectable in 29/117 patients (24.8%) and was associated with inferior RFS (HR = 2.42; 95% CI = 1.11–5.27; *p* = 0.022) and inferior OS (HR = 5.54; 95% CI = 1.01–30.35; *p* = 0.026) ✓ ctDNA was detected in 12/116of the first postsurgical samples (10.3%) and was associated with shorter RFS (HR = 3.04; 95% CI = 1.22–7.58; *p* = 0.012) ✓ Longitudinal ctDNA-positive patients (37/119; 31.1%) had shorter RFS (HR, 3.46; 95% CI, 1.59–7.55; *p* < 0.001) and shorter OS (HR = 9.99; 95% CI = 1.17–85.78; *p* = 0.010) in comparison to longitudinal ctDNA-negative patients	8.71
[138] (2022)	21	IA–IIIB	NGS for an18-gene panel/Pro	26.2	♦ Before surgery ♦ During surgery ♦ 1–2 weeks post-surgery ♦ Longitudinal detection	✓ Proportion of patients who were ctDNA-positive before surgery: 57% (12/21) ✓ ctDNA detection rates and ctDNA concentrations were significantly higher in plasma obtained during surgery compared to preoperative specimens (57% vs. 19%; 12.47 ng/mL vs. 6.64 ng/mL) ✓ Positive ctDNA detection in early postoperative plasma samples was associated with shorter RFS (*p* = 0.013) and OS (*p* = 0.004)	10.31
[139] (2022)	330	I–III	NGS for a 769-gene panel/Pro	35.6	♦ 1 week before surgery ♦ 3 days after surgery ♦ 1 month after surgery	✓ Preoperative ctDNA positivity was associated with lower RFS (HR = 4.2; *p* < 0.001) ✓ The presence of MRD (ctDNA positivity at 3 days and/or 1 month post-surgery) was a strong predictor for disease relapse (HR = 11.1; *p* < 0.001) ✓ MRD-positive patients who received adjuvant therapies had improved RFS compared to those who did not receive adjuvant therapy (HR = 0.3; *p* = 0.008), whereas MRD-negative patients who received adjuvant therapies had lower RFS compared to those who did not receive adjuvant therapy (HR = 3.1; *p* < 0.001)	NR
[140] (2022)	88	IA–IIIB	RaDaRTMNGS	36	♦ Before treatment ♦ During treatment ♦ After the end of treatment ♦ Longitudinal detection ♦ Treatment: surgery (*n* = 61); surgery + adjuvant chemotherapy/radiotherapy (*n* = 8); chemoradiotherapy (*n* = 19)	✓ Proportion of patients who were ctDNA− positive before treatment: 51% (40/78) I: 24% (10/41); II: 77% (17/22); III: 87% (13/15) ✓ ctDNA was detected after treatment in 18/28 (64.3%) of patients who demonstrated a clinical recurrence of their primary tumor ✓ Detection within the landmark timepoint of 2 weeks 4 months after the end of treatment occurred in 17% of patients and was associated with shorter RFS (HR = 14.8, *p* < 0.00001) and OS (HR = 5.48, *p* < 0.0003) ✓ ctDNA was detected 1–3 days after surgery in 25% of patients and was not associated with disease recurrence ✓ Preoperative detection was associated with shorter OS (HR = 2.97; *p* = 0.01) and RFS (HR = 3.14, *p* = 0.003)	7.08

## Data Availability

Not applicable.

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
