# Peer review of "Can Liquid Biopsy Based on ctDNA/cfDNA Replace Tissue Biopsy for the Precision Treatment of EGFR-Mutated NSCLC?"

_jcm, 2023, doi:10.3390/jcm12041438_

Round 1
Reviewer 1 Report
Dear Authors,
Thank you for an interesting review. Please find attached my comments and questions.

Reviewer 2 Report
These studies are conducted by various specimen samples and laboratory procedure, do you think that gene abnormality should be detected using the same method, if you reflect the result of these studies for treatment?
Reviewer 3 Report
The review describes the possibilities of detecting genetic markers of the disease using a liquid biopsy. Data with different sensitivity and specificity are presented. The absence of reference intervals for such methods becomes clear. However, the presented review shows the prospects for the use of such markers for minimally residual tumors (MRD).
Reviewer 4 Report
The article from Li et al proposes a review of the role of liquid biopsy in management of NSCLC, focusing on EGFR mutations. They discuss the aspects of initial detection, identification of resistance mechanisms and monitoring of minimal residual disease.
The title is interesting but at the end of the manuscript, it is unsure that the elements of the manuscript allow answering to the proposed entitled question. One fundamental question is why the focus is on EGFR only, as the current management of lung cancer embrace evaluation of EGFR along many other biomarkers, some being considered as a required minimal testing (EGFR, ALK, ROS1 by last CAP/IASLC/AMP guidelines)? So, while the review is interesting and discuss some fundamentals elements of liquid biopsy regarding to EGFR testing, the relevance of looking at only EGFR is limited and there is often unclear distinction between studies focused on EGFR only vs those that include EGFR amongst comprehensive molecular testing. Indeed, it is unclear in several sections if the reported findings pertain to biomarkers overall, oncogenic drivers or activating mutations in EGFR, especially at the beginning. The discussion focuses on some technical aspects of selected studies or clinical outcomes, evacuating the global perspective of EGFR amongst multiple other biomarkers that are clinically significant and also overlapping clinical and research applications. Thus, a better organisation of the underlying ideas would improve what information the authors want to relay.
A thorough review of syntax and wording would be necessary as it sometimes obscures the message and impairs the flow of ideas:
Per examples
-line 29: but most patients with advanced NSCLC have difficulty in obtaining tumor tissue
line 39: refers to the substrate used in liquid biopsy, not to «EGFR» liquid biopsy (?); such statement would also require some references, especially for uncommon substrates.
Figure 1 legend: a more neutral wording would be more appropriated in a scientific review («Great potential»). Also, the choice of making a figure including all potential «liquid biopsy» sources contrast with the relatively modest discussion of sources other than blood in the manuscript, essentially limited to the end.
Line 56: individualized and precise targeted therapy: the term «precision medicine» or «targeted therapy» is more commonly used.
Line 60: «came into being» ?
Line 100: the treatment of Gefitinib
Line 280
Section 2: heading refers to EGFR but it is not clear in the first parapgraph if authors refer specifically to EGFR mutations or actionable alterations in NSCLC overall (highly likely with the numbers provided). Then it leads to the description of the findings of a paper (ref 19), still not restrained to EGFR findings, that leads to a recommendation/suggestion from the authors - is that the role of a review to compare the results of different studies before to lead to a recommendation ? This paragraph specifically, and generally the manuscript, would benefit from a better distinction of ideas using shorter paragraphs.
Line 112-113: this seems like an interpretation from the authors, they should clarify ? It seems incorrect as other clinical factors such as clinical extension of disease could be associated with a higher likelihood of a positive liquid biopsy result and be associated with worse prognosis, not the result of a liquid biopsy assay.
Line 192: the authors state that low expression of EGFR is a primary resistance mechanism, but that seems odd - should clarify and cite appropriated reference.
Lines 197 and below. This is a very short review of acquired resistance mechanisms with emphasis on targets rather than mechanisms; how these mechanisms affect the selection of an appropriate liquid biopsy assay for AR to EGFR TKIs (p.ex. fusion detection) is not discussed. How the assays (NGS with or without fusion capture; NGS is not enough in that kind of review) have allowed proper coverage of all these mechanisms is not clear from Table 2. In contrast, the number of genes from the NGS panels are included in Table 3.
Paragraph 280: no mention about the allelic configuration of T790M and C797S mutations, a key aspect on this topic clinically.
Round 2
Reviewer 4 Report
N/A